# Learning Functions on Multiple Sets using Multi-Set Transformers

**Kira A. Selby**[1,2]    **Ahmad Rashid**[1,2,3]    **Ivan Kobyzev**[3]    **Mehdi Rezagholizadeh**[3]    **Pascal Poupart**[1,2]

[1]Cheriton School of Computer Science, University of Waterloo, Waterloo, Ontario, Canada
[2]Vector Institute , Toronto, Ontario, Canada
[3]Huawei Noah's Ark Lab, Montreal, Quebec, Canada

## Abstract

We propose a general deep architecture for learning functions on multiple permutation-invariant sets. We also show how to generalize this architecture to sets of elements of any dimension by dimension equivariance. We demonstrate that our architecture is a universal approximator of these functions, and show superior results to existing methods on a variety of tasks including counting tasks, alignment tasks, distinguishability tasks and statistical distance measurements. This last task is quite important in Machine Learning. Although our approach is quite general, we demonstrate that it can generate approximate estimates of KL divergence and mutual information that are more accurate than previous techniques that are specifically designed to approximate those statistical distances.

## 1 INTRODUCTION

Typical deep learning algorithms are constrained to operate on either fixed-dimensional vectors or ordered sequences of such vectors. This is a limiting assumption which prevents the application of neural methods to many problems, particularly those involving sets. While some investigation has now been done into the problem of applying deep learning to functions on sets [Lee et al., 2019, Zaheer et al., 2017], these works all focus on the problem of learning a function on a single input set to either a corresponding set of outputs or a single fixed-dimensional output. Very little work has been done on functions of *multiple* sets.

This work seeks to fill that gap. We propose a general architecture to learn functions on multiple permutation invariant sets based on previous works by Zaheer et al. [2017] and Lee et al. [2019].[1] We demonstrate that this architecture is a universal approximator of *partially-permutation-invariant* functions, based on the work of Yun et al. [2019]. We demon-

strate how this architecture vastly outperforms existing approaches and several baselines on a variety of tasks, as well as discuss how this can be applied to the special case of learning distance functions between two sets. This latter application allows this model to be trained as an effective estimator of quantities such as KL divergence and Mutual Information, which are highly relevant for many applications within machine learning. Furthermore, we demonstrate how to obtain an architecture that generalizes with the dimensionality of the data by making the computation equivariant with respect to the input dimensions.

## 2 RELATED WORK

Our method is based on the work of Zaheer et al. [2017], who originally drew attention to the problem of using neural networks to approximate functions on permutation-invariant sets. We draw particularly from the work of Lee et al. [2019], who extended this work to use transformer-based models on sets.

Gui et al. [2021] also address the idea of designing neural networks to learn functions between multiple permutation-invariant sets. Their work, however, focuses on graph embeddings as a primary application, and does not consider estimating distances or divergences. Their method also relies on a more simplistic architecture that has been criticized in Wagstaff et al. [2019a], whereas our proposed architecture has a number of theoretical and empirical advantages.

## 3 METHOD

### 3.1 BACKGROUND

Consider first the problem of learning a function upon a single set $X = \{x_1, ..., x_n\}$ in $\mathbb{R}^d$. As discussed in the

---

[1]https://github.com/krylea/partial-invariance

*Accepted for the 38[th] Conference on Uncertainty in Artificial Intelligence*  (UAI 2022).

work of Zaheer et al. [2017], this general problem takes on one of two forms: the *permutation-equivariant* and -*invariant* cases. In the permutation-equivariant case, the function takes the form $f : 2^{\mathbb{R}^d} \to 2^{\mathbb{R}^{d'}}$, and must obey the restriction that permutations of the inputs correspond to identical permutations of the outputs - i.e. for a permutation $\pi$,

$$f(\pi(X)) = \pi(f(X))$$

In the permutation-invariant case, the function takes the form $f : 2^{\mathbb{R}^d} \to \mathbb{R}^{d'}$, and must obey the restriction that permutations of the inputs correspond to no change in the output - i.e.

$$f(\pi(X)) = f(X)$$

Zaheer et al. [2017] proposed an architecture to learn such functions that we will refer to as the *sum-decomposition* architecture (following Wagstaff et al. [2019a]). This architecture proposes to learn permutation-invariant functions using the model:

$$f(X) = \rho\left(\sum_{x \in X} \phi(x)\right) \qquad (1)$$

Each member of the set $x$ is encoded into a latent representation $\phi(x)$, which are then summed and decoded to produce an output. While this architecture is adequate for some purposes, it is also very simple, and has difficulty modeling interactions between multiple elements in the set. In particular, Wagstaff et al. [2019b] proved in their paper that *sum-decomposable* architectures such as this require each element to be mapped to a latent vector with latent size at least as large as the maximum number of elements in the input set in order to be universal approximators of functions on sets. Practically speaking, this is a major restriction when working with large sets, because this introduces a hard maximum on the size of the input set for a given latent size.

Lee et al. [2019] improved on this architecture in their paper, proposing an architecture they referred to as 'Set Transformers'. Based on their architecture, we can present a general model for attention-based models on sets as follows. Let MHA$(X, Y)$ be the standard multiheaded attention operation defined in Vaswani et al. [2017] with queries $X$ and keys/values $Y$. We define a single *transformer block* by:

$$\begin{aligned}
\text{ATTN}(X) &= \text{LN}(X + \text{MHA}(X, X)) \\
T(X) &= \text{LN}(\text{ATTN}(X) + \text{FF}(\text{ATTN}(X)))
\end{aligned} \qquad (2)$$

wherein LN is the Layer Norm operation [Ba et al., 2016], and FF is a 2-layer feedforward network with ReLU activation applied independently to each element in the set. Note that this transformer block lacks positional encodings, and is thus a permutation-equivariant operation.

In order to create a general set transformer model, we can stack multiple of these blocks followed by a pooling opera-

tion $\Gamma$ and a decoder $\rho$ to obtain:

$$f(X) = \rho\left(\Gamma_X T_l(T_{l-1}(...T_1(X)))\right) \qquad (3)$$

The Set Transformer architecture bears some similarities to another model proposed by Santoro et al. [2017]. They propose a simpler permutation-invariant architecture called "relation networks", which follow the model:

$$f(X) = \rho\left(\sum_{x_i \in X} \sum_{x_j \in X} \theta(x_i, x_j)\right) \qquad (4)$$

wherein $\rho$ is again a decoder, and $\theta$ is a feedforward *pairwise* encoder which encodes the relationship between each pair of elements in $X$.

In general, we can consider all three of these architectures to consist of an equivariant encoder $\phi$ on the set $X$, followed by a pooling operation $\Gamma$ and decoder $\rho$:

$$f(X) = \rho\left(\Gamma_X \phi(X, X)\right) \qquad (5)$$

In both the Set Transformer and Relation Network case, $\phi(X, X)$ can be written as

$$\phi(X, X) = \Lambda_X \theta(X, X) \qquad (6)$$

where $\theta(X, X)$ is a pairwise encoder that computes an encoding of each pair $(x_i, x_j)$, then $\Lambda$ is a form of pooling operation that reduces this $N \times N$ encoding matrix into a single vector for each element in $X$. For the base relation network architecture, $\theta$ is simply a feedforward neural net, and both pooling operations take the form of sums. For the set transformer architecture, $\phi(X, X)$ is the multiheaded self-attention operator, with $\theta(X, X)$ being the dot product of the transformed queries and keys and $\Lambda$ consisting of a softmax and matrix multiplication by the transformed value matrix. The pooling operator $\Gamma$ is given by the pooling-by-multiheaded-attention (PMA) operator defined in Lee et al. [2019].

## 3.2 MULTIPLE SETS

We propose to extend these methods to the case of multiple permutation invariant sets - which we refer to as the case of *partial permutation invariance*. We say a function $f : 2^{\mathbb{R}^d} \times 2^{\mathbb{R}^d} \to \mathbb{R}^{d'}$ is *partially permutation invariant* if $\forall \pi_1, \pi_2$ it obeys the property:

$$f(\pi_1(X), \pi_2(Y)) = f(X, Y)$$

Similarly, we say a function $f : 2^{\mathbb{R}^d} \times 2^{\mathbb{R}^d} \to 2^{\mathbb{R}^d} \times 2^{\mathbb{R}^d}$ is *partially permutation equivariant* if it obeys the property:

$$f(\pi_1(X), \pi_2(Y)) = (\pi_1(f_X(X, Y)), \pi_2(f_Y(X, Y)))$$

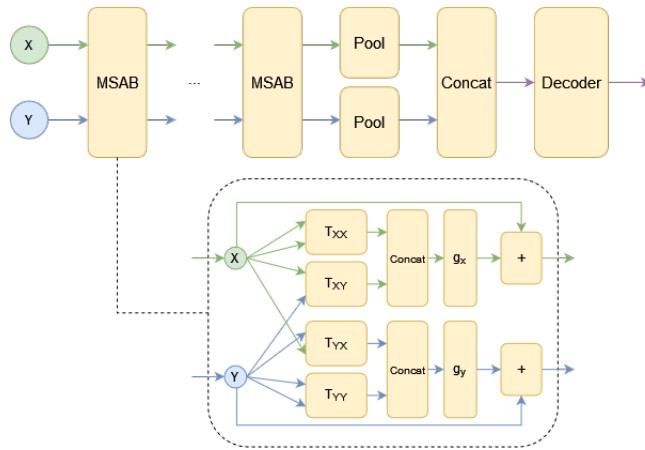

Figure 1: Diagram of the Multi-Set Transformer and Multi-Set Attention Block

Gui et al. [2021] propose a similar definition in their work, where they define a partially permutation invariant model:

$$f(X_1, ..., X_m) = \rho \left( \sum_{x \in X_1} \phi_1(x), ..., \sum_{x \in X_m} \phi_m(x) \right) \quad (7)$$

As mentioned previously, this sum-decomposition architecture has a number of limitations, and instead we choose to follow the method of Lee et al. [2019]. This architecture is also advantageous for other reasons related specifically to the modelling of distance functions. Models such as transformers which explicitly model the relationship between pairs of elements in a set carry a useful inductive bias for learning distance functions. Consider the case of computing the Wasserstein distance, wherein computing the ground distance (e.g. euclidean distance) between every pair of elements of the sets is a necessary step. Similarly for quantities such as KL divergence, methods such as the algorithm of Wang et al. [2009] often rely on nearest-neighbour distances as a useful proxy for the concentration of the distribution.

### 3.3   OUR MODEL

We choose to instead begin from the model presented in Equation 5. In order to generalize this, let us now consider applying these architectures to the case where the single input $X$ is now replaced by $X \bigsqcup Y$ - the concatenation of the two inputs $X$ and $Y$. When the encoder acts upon this input, $\phi(X, X)$ is replaced by:

$$\phi(X \bigsqcup Y, X \bigsqcup Y) = \prod_{X \bigsqcup Y} \begin{pmatrix} \theta(X, X) & \theta(X, Y) \\ \theta(Y, X) & \theta(Y, Y) \end{pmatrix} \quad (8)$$

Instead of having a single encoder learn these four relationships, we can split $\theta$ into four separate pair encoders: $\theta_{XX}$, $\theta_{XY}$, $\theta_{YX}$, and $\theta_{YY}$. Rather than pooling over the entirety

of the joint set $X \bigsqcup Y$, we pool over only the first input:

$$\phi_{xx}(X, X) = \prod_X \theta_{xx}(X, X)$$
$$\phi_{xy}(X, Y) = \prod_X \theta_{xy}(X, Y)$$
$$\phi_{yx}(Y, X) = \prod_Y \theta_{yx}(Y, X)$$
$$\phi_{yy}(Y, Y) = \prod_Y \theta_{yy}(Y, Y)$$

In this manner, each encoder learns the relationships between one of the four pairs of sets separately. This information can then be recombined to produce output encodings:

$$\phi_x(X, Y) = g_x(\phi_{xx}(X, X), \phi_{xy}(X, Y))$$
$$\phi_y(X, Y) = g_y(\phi_{yx}(Y, X), \phi_{yy}(Y, Y))$$

Equation 5 then becomes

$$f(X, Y) = \rho \left( \prod_X \phi_x(X, Y), \prod_Y \phi_y(X, Y) \right) \quad (9)$$

This structure satisfies the property of partial permutation equivariance, and allows the model to retain the benefits of explicitly representing relationships between each pair of elements and each pair of sets. This general model can now be used to extend any single-set model defined by Equation 5 - including both Set Transformers and Relation Networks.

### 3.4   MULTI-SET TRANSFORMER

The primary architecture we consider is the *multi-set transformer* architecture, which follows from constructing the model defined in Equation 9 with transformer encoders as $\phi$. We define the *multi-set attention block* $\text{MSAB}(X, Y) = (Z_X, Z_Y)$ where

$$Z_X = X + g_x(T_{xx}(X, X), T_{xy}(X, Y)) \quad (10)$$
$$Z_Y = Y + g_y(T_{yx}(Y, X), T_{yy}(Y, Y)) \quad (11)$$

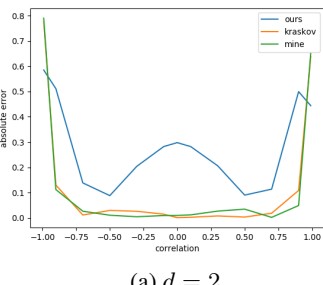 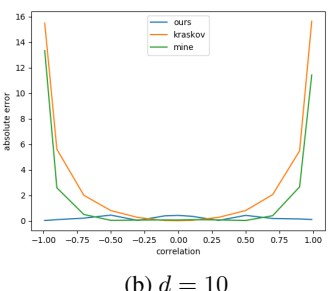 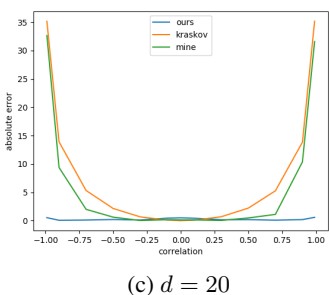

(a) $d = 2$  (b) $d = 10$  (c) $d = 20$

Figure 2: Plot of absolute error in predicted mutual information for correlated Gaussian data with 2d, 10d and 20d marginals for our model and baselines.

and where $T_{ab}(A, B)$ is a transformer block as defined in Eq. 2, and the functions $g$ are 1-layer feedforward networks with ReLU activations which are applied to the element-wise concatenation of the outputs of the two transformer blocks. These MSABs can now be treated like regular transformer blocks and stacked to form a deep encoder. Figure 1 illustrates our model with MSABs. We can then define a multi-set analogue of Eq. 3:

$$f(X, Y) = \rho \left( \Gamma_X \phi(X, Y)_X, \Gamma_Y \phi(X, Y)_Y \right) \quad (12)$$

$\phi$ is an encoder formed of stacked MSABs, which produces outputs of $Z_X$ and $Z_Y$. These outputs are then pooled over $X$ and $Y$ independently, concatenated, then passed into a feedforward decoder to produce the final output. See Section 1 of the supplementary material for a detailed discussion of how our multi-set attention model is derived from a single-set attention block.

## 3.5 VARIABLE-DIMENSION ENCODERS

Another application of permutation-invariance that will be particularly useful when discussing distance functions is invariance to the input dimension itself. Applying the principles of permutation invariance to the input dimension itself, we can arrive at a formulation where the model is in fact invariant to the input dimension, and can accept inputs of any size. A traditional machine learning pipeline in - for example - NLP might learn an embedding scheme in which different dimensions encode different semantic representations of the input sequence, and should thus be treated differently from one another. In order to compute a function such as the Wasserstein distance or KL divergence on a diverse array of distributions, however, each dimension must be treated symmetrically. In the setting of statistical distances, this representation of the input dimension carries an inductive bias that is useful for the model, and generally leads to improved performance.

Zaheer et al. [2017] propose a simplistic form of this in their original paper. They demonstrate that a linear layer with a weight matrix that takes the form $\Theta = \lambda I + \gamma 11^T$ is equivariant with respect to the input dimension. This essentially corresponds to a neural network that computes:

$$y_i = \lambda x_i + \gamma \sum_i x_i$$

Each output is thus computed from a constant multiple of the corresponding input added to a multiple of the sum of all inputs. This has some unfortunate properties, however, since it is constrained to an output size that is exactly equal to the input size at every layer. The solution to this is to introduce multiple channels. Instead of mapping each input dimension to a single output dimension, each input dimension can be mapped to a multichannel output. Multiple encoder layers can thus be stacked, each acting only on this multichannel representation of the input dimensions, and treating the input dimension itself as a batch dimension. Then, after these encoder layers are applied, a pooling operation can be introduced over the input dimension to obtain a fixed dimensional output. This procedure allows inputs of any size to be mapped to a fixed dimension encoding.

In accordance with this, we propose an analogous *multi-channel transformer block*, wherein the weight matrices are applied similarly as multichannel transformations which treat the input dimension as a batch dimension. A standard multiheaded attention block receives inputs $X, Y \in \mathbb{R}^{n \times d}$ and computes:

$$\text{MHA}(X, Y) = \sigma \left( (XW_Q)(YW_K)^T \right) YW_V W_O \quad (13)$$

Our multichannel attention block instead is a function from $\mathbb{R}^{n \times d \times n_c} \times \mathbb{R}^{m \times d \times n_c}$ to $\mathbb{R}^{n \times d \times n_c}$ which computes:

$$A = \sigma \left( \sum_{i=1}^{d} (X_{:,i}W_Q)(Y_{:,i}W_K)^T \right)$$
$$\text{MHA}(X, Y)_{:,i} = AY_{:,i}W_V W_O$$

Our multichannel transformer architecture consists of an initial projection from a single input channel up to $k$ input

channels, followed by $n_b$ multichannel transformer blocks, followed by a max pooling over the input dimension $d$. This is also compatible with the Multi-Set Attention Block architecture, in which case the multichannel transformer blocks are replaced by multichannel MSABs, with a max pooling at the end as before.

# 4 THEORETICAL ANALYSIS

We demonstrate that our proposed multi-set transformer architecture is a universal approximator on *partially permutation equivariant* functions, and that combined with a pooling layer it is also a universal approximator of *partially permutation invariant* functions.

First, some preliminaries. We will follow the notation in Yun et al. [2019], for their Theorem 2 forms a foundation for the theorem we will state shortly. Let $\mathcal{F}_{PE}$ be the class of all continuous permutation-equivariant functions with compact support from $\mathbb{R}^{d \times n}$ to $\mathbb{R}^{d \times n}$. Given $f, g : \mathbb{R}^{d \times n} \to \mathbb{R}^{d \times n}$ and $1 \leq p \leq \infty$, let

$$d_p(f, g) = \left( \int \|f(\mathbf{X}) - g(\mathbf{X})\|_p^p d\mathbf{X} \right)^{1/p} \quad (14)$$

Let $t^{h,m,r} : \mathbb{R}^{d \times n} \to \mathbb{R}^{d \times n}$ denote a transformer block with an attention layer with $h$ heads of size $m$ and a feedforward layer with $r$ hidden nodes. Then, let $\mathcal{T}^{h,m,r}$ define the class of functions $g : \mathbb{R}^{d \times n} \to \mathbb{R}^{d \times n}$ such that $g$ consists of a composition of transformer blocks of the form $t^{h,m,r}$. We can now restate Theorem 2 from Yun et al. [2019], which is given by:

**Theorem 4.1.** *Let $1 \leq p \leq \infty$ and $\epsilon > 0$, then for any given $f \in \mathcal{F}_{PE}$ there exists a transformer network $g \in \mathcal{T}^{2,1,4}$ such that $d_p(f, g) \leq \epsilon$. [Yun et al., 2019]*

To extend this to the case of partial permutation equivariance, we must now modify these definitions slightly. Let $\mathcal{F}_{PPE}$ be the class of all continuous partially-permutation-equivariant functions on two sets with compact support from $\mathbb{R}^{d \times n} \times \mathbb{R}^{d \times m}$ to $\mathbb{R}^{d \times n} \times \mathbb{R}^{d \times m}$. Let $c^{h,m,r} : \mathbb{R}^{d \times n} \times \mathbb{R}^{d \times m} \to \mathbb{R}^{d \times n} \times \mathbb{R}^{d \times m}$ denote a multi-set attention block with attention layers with $h$ heads of size $m$ and feedforward layers with $r$ hidden nodes, and let $\mathcal{T}_C^{h,m,r}$ define the class of functions $g : \mathbb{R}^{d \times n} \times \mathbb{R}^{d \times m} \to \mathbb{R}^{d \times n} \times \mathbb{R}^{d \times m}$ such that $g$ consists of a composition of MSABs of the form $c^{h,m,r}$. Our theorem now states:

**Theorem 4.2.** *Let $1 \leq p \leq \infty$ and $\epsilon > 0$, then for any given $f \in \mathcal{F}_{PPE}$ there exists a multi-set transformer network $g \in \mathcal{T}_C^{2,2,4}$ such that $d_p(f, g) \leq \epsilon$.*

A proof of Theorem 4.2 is given in Section 3 of the supplementary material. If we define $\mathcal{F}_{PPI}$ to be the class of all continuous partially-permutation-invariant functions on two

sets with compact support from $\mathbb{R}^{d \times n} \times \mathbb{R}^{d \times m}$ to $\mathbb{R}^d$, this now directly leads to the corollary:

**Corollary 4.3.** *Let $1 \leq p \leq \infty$ and $\epsilon > 0$, then for any given $f \in \mathcal{F}_{PPI}$ there exists a function $f(X, Y) = \max g(X, Y)$ such that $d_p(f, g) \leq \epsilon$, wherein $g \in \mathcal{T}_C^{2,2,4}$ is a multi-set transformer network.*

Observe that this corollary follows directly from Theorem 4.2, since for any function $f \in \mathcal{F}_{PPI}$ we can construct $g \in \mathcal{F}_{PPE}$ such that $g(X, Y)_{:,j} = f(X, Y) \ \forall j$ (i.e., a set of outputs where each entry simply contains $f(X, Y)$). $f$ thus obeys the equation $f(X, Y) = \max_j g(X, Y)_{:,j}$. Therefore, any function in $f \in \mathcal{F}_{PPI}$ can be expressed as a pooling function applied to a function $g \in \mathcal{F}_{PPE}$ and $f$ can thus be approximated by a multi-set transformer network by simply approximating $g$ as per Theorem 4.2.

# 5 EXPERIMENTS

In order to evaluate the model, we consider several tasks, including a number of simple image-based set tasks as well as the aforementioned distance functions. We compare our model against the PINE model proposed in [Gui et al., 2021], as well as a number of single-set models. For those baselines, we take a single-set architecture such as Deep Sets (Single-Set RFF), Relation Networks (Single-Set RN) or Set Transformers (Single-Set Transformer), compute pooled representations for each of $X$ and $Y$, then concatenate these representations and pass them into a feedforward decoder. Finally, we also compare to a simple transformer baseline wherein a Set Transformer is applied to the union $X \bigsqcup Y$ (Union Transformer).

We also consider several variants and ablations of our model. The two variants of our model include Multi-Set Transformer and Multi-Set RN (Relation Networks). In the latter, transformer blocks are replaced by relation network blocks for the four encoders, with max pooling operations for both $\Lambda$ and $\Gamma$. Then, we also consider several ablations of our model. First, we consider a variant where $g_x(T_{xx}(X, X), T_{xy}(X, Y)) = T_{xx}(X, X) + T_{xy}(X, Y)$ (referred to as Sum-Merge). Second, we consider modifications to the four-block encoder structure itself by removing $T_{XX}$ and $T_{YY}$ - leaving only the cross terms $T_{XY}$ and $T_{YX}$ (referred as Cross-Only). Finally, the single set transformer baseline (Single-Set Transformer) can itself be considered an ablation of our model with the cross-set blocks removed instead of the same-set blocks. For all experiments we perform three trials and report the average and standard deviation.

Hyperparameter settings for all experiments and other details can be found in Section 2 of the supplementary material.

|  | d=2 | d=4 | d=8 | d=16 |
|---|---|---|---|---|
| **Baselines** | | | | |
| KNN | 0.2047 | 0.5662 | 4.0584 | 28.0382 |
| PINE | $0.1737 \pm 0.0003$ | $0.4958 \pm 0.0011$ | $2.0804 \pm 0.0004$ | $10.534 \pm 0.0109$ |
| Single-Set RFF | $0.1219 \pm 0.0114$ | $0.4400 \pm 0.0159$ | $1.7770 \pm 0.0119$ | $8.0078 \pm 0.1636$ |
| Single-Set RN | $0.1555 \pm 0.0007$ | $0.5264 \pm 0.0019$ | $2.1425 \pm 0.0032$ | $9.5963 \pm 1.6214$ |
| Single-Set ST | $0.0732 \pm 0.0032$ | $0.2601 \pm 0.0043$ | $1.6885 \pm 0.0337$ | $7.5911 \pm 0.2125$ |
| Union Transformer | $0.1747 \pm 0.0004$ | $0.4990 \pm 0.0006$ | $2.2665 \pm 0.0006$ | $9.7316 \pm 0.0457$ |
| **Our Model** | | | | |
| Multi-Set Transformer | $0.0731 \pm 0.0011$ | $\mathbf{0.1903} \pm 0.0082$ | $0.9212 \pm 0.0186$ | $11.105 \pm 0.0717$ |
| Multi-Set RN | $0.1061 \pm 0.0020$ | $0.3926 \pm 0.0144$ | $1.5170 \pm 0.0430$ | $7.4002 \pm 0.1263$ |
| Cross-Only | $0.0792 \pm 0.0019$ | $0.1968 \pm 0.0014$ | $0.9926 \pm 0.0351$ | $5.4214 \pm 0.1694$ |
| Sum-Merge | $\mathbf{0.0699} \pm 0.0008$ | $0.1953 \pm 0.0047$ | $0.9320 \pm 0.0138$ | $10.5080 \pm 0.0662$ |
| Multi-Set-Transformer-Equi | $0.0726 \pm 0.0017$ | $0.1917 \pm 0.0020$ | $\mathbf{0.8002} \pm 0.0222$ | $\mathbf{4.7000} \pm 0.2966$ |

Table 1: Mean absolute error of models trained on Gaussian mixture data for estimating KL divergence.

## 5.1 STATISTICAL DISTANCES

One particular application of partially-permutation-invariant models that is worth highlighting is their ability to learn to approximate statistical distances between distributions such as the KL divergence or mutual information. Both Mutual Information and the KL divergence are useful metrics that are widely used in a variety of settings within machine learning, and both are very difficult to estimate for any but the simplest distributions. We consider both our proposed model, as well as the dimension-equivariant model discussed in Sect. 3.5.

### 5.1.1 KL Divergence

Training the estimator to learn the KL divergence has unique challenges, as calculating the ground truth requires the log likelihood for both the source and target distributions. In order to train our model to learn the KL divergence between a general class of distributions, we need to find a class of models that are effective universal approximators and also admit a tractable log likelihood. The most obvious class of models fitting this criteria is that of Gaussian mixture models. We generate Gaussian mixtures with a uniformly random number of components (between 1 and 10) and mixture weights sampled from a uniform Dirichlet distribution. The means of each Gaussian are generated from a uniform distribution, and the covariance matrices are generated by multiplying a correlation matrix sampled from a Lewandowski-Kurowicka-Joe (LKJ) distribution (with $\nu = 5$) by a vector of covariances distributed according to a log-normal distribution (with $\mu_0 = 0, \sigma_0 = 0.3$).

Each training example consists of a random number of points $X \sim p_X$ and a random number of points $Y \sim p_Y$ (with $N_X, N_Y \in [100, 150]$). The ground truth is estimated by a Monte Carlo estimate of the true KL divergence using

the closed-form log likelihoods, with the generated points $X$ as the samples. We normalize the generated data by computing the mean and covariance across both $X$ and $Y$, then applying a whitening transformation

$$[X'; Y'] = \Sigma_{XY}^{-1/2} \left([X; Y] - \mu_{XY}\right) \tag{15}$$

under which the KL divergence is invariant.

We compare our model (with and without dimension-equivariance) against the aforementioned baselines, as well as the k-nearest-neighbours estimator of Wang et al. [2009]. Table 1 shows the mean average error of each model on Gaussian mixture data, averaged over 3 runs. Our model has the lowest error on all dimensions considered. The dimension-equivariant model performed approximately equal to the standard transformer model in low dimension, but performed significantly better in high dimension.

Convergence was a significant issue with the GMM data in higher dimensions, since as the dimensionality increased the true KL divergence of the generated distributions would often explode. This effect was especially notable when the concentration parameter of the LKJ distribution was small, but always occurs once the dimensionality gets large enough.

### 5.1.2 Mutual Information

We also show the effectiveness of our method for estimating mutual information. Following previous work [Belghazi et al., 2018, Kraskov et al., 2004], we use Gaussians with componentwise correlations $\rho \in (-1, 1)$, with standardized Gaussian marginals. Training examples are generated in a similar fashion as in the KL case, with a $\rho$ sampled uniformly from the interval $(-1, 1)$, then a random number of samples between 100 and 150 drawn from the resulting distribution for each of X and Y. We plot the performance of our model for varying values of $\rho$ compared to both the

| | Omniglot | | MNIST | |
| Model | Acc | L1 | Acc | L1 |
|---|---|---|---|---|
| **Baselines** | | | | |
| Pine | $0.6618 \pm 0.0133$ | $0.8237 \pm 0.0056$ | $0.4682 \pm 0.0039$ | $1.2438 \pm 0.0413$ |
| Single-Set RFF | $0.6310 \pm 0.0021$ | $0.8915 \pm 0.0424$ | $0.4421 \pm 0.0122$ | $1.3633 \pm 0.0282$ |
| Single-Set RN | $0.6724 \pm 0.0059$ | $0.8110 \pm 0.0094$ | $0.5369 \pm 0.0977$ | $1.0971 \pm 0.2182$ |
| Single-Set Transformer | $0.7242 \pm 0.0031$ | $0.7329 \pm 0.0056$ | $0.9123 \pm 0.0338$ | $0.4664 \pm 0.0622$ |
| Union Transformer | $0.6296 \pm 0.0009$ | $0.8680 \pm 0.0094$ | $0.5339 \pm 0.0034$ | $1.1110 \pm 0.0055$ |
| **Our Models** | | | | |
| Multi-Set Transformer | $0.8538 \pm 0.0077$ | $0.5431 \pm 0.0115$ | $0.9746 \pm 0.0038$ | $0.3136 \pm 0.0124$ |
| Multi-Set RN | $\mathbf{0.8699} \pm 0.0048$ | $\mathbf{0.5166} \pm 0.0076$ | $0.9782 \pm 0.0104$ | $0.3184 \pm 0.0564$ |
| Cross-Only | $0.8189 \pm 0.0342$ | $0.5929 \pm 0.0516$ | $0.9723 \pm 0.0015$ | $0.3128 \pm 0.0109$ |
| Sum-Merge | $0.8544 \pm 0.0071$ | $0.5416 \pm 0.0080$ | $\mathbf{0.9784} \pm 0.0011$ | $\mathbf{0.2600} \pm 0.0151$ |

Table 2: Average accuracy and L1 error of each model on the MNIST and Omniglot counting tasks across 3 runs (higher is better for accuracy and lower for L1).

Kraskov et al. [2004] and MINE [Belghazi et al., 2018] baselines in 2, 10 and 20 dimensions (see Fig. 2). Our model performs somewhat worse than the other methods shown in the 2-dimensional case, but is almost indistinguishable from the ground truth in the 10 and 20-dimensional cases. Note also that while methods such as MINE must be trained on a particular dataset in order to predict its mutual information, our method need only be trained once, and can then be used for inference on any similar dataset wthout retraining.

## 5.2 IMAGE TASKS

We begin by looking at a selection of tasks similar to those considered by Lee et al. [2019] and Zaheer et al. [2017]. We study the model's ability to perform a number of simple set-based operations between sets of images. When working with image or text data, each example was first individually encoded as a fixed-size vector using an appropriate image or text encoder, then passed through the set based model.

### 5.2.1 Counting Unique Images

For the first task, the models were given input sets consisting of images of characters. The task was to identify the number of unique characters that were shared between the two input sets of a variable number of images drawn from the MNIST and Omniglot [Lake et al., 2015] datasets (6-10 images for Omniglot, 10-30 images for MNIST). For this task, we used simple CNN encoders that were pretrained on the input datasets as classifiers for a short number of steps, then trained end to end with the set-based model. We found that our models convincingly outperformed the alternatives - achieving almost 98% accuracy on the MNIST task and 83-85% accuracy on the Omniglot task (see Table 2). In each case, our models outperformed the baselines by considerable margins. The RN-based model outperformed the

transformer model by a margin of about 1.5% on the Omniglot task, and they performed equivalently on the MNIST task. The ablations performed largely similarly to the base model, with degraded performance only in the case of the Cross-Only model on the Omniglot dataset.

### 5.2.2 Alignment

While the first task was purely synthetic, this second task is representative of a general class of applications for this model - predicting alignment between two sets. The first example of this we chose was image captioning on the MSCOCO dataset. The models were given a set of 8-15 images and a set of captions of the same size, and tasked to predict the probability that the two sets were *aligned* - i.e. that the given set of captions consisted of captions for the given set of images. For this task, we used the pretrained models BERT and ResNet-101 as encoders for the text and images respectively. For the second example, we chose to use cross-lingual embeddings. Lample et al. [2018] show that there is a geometric relationship between learned FastText embeddings across languages. As such, the model should be able to predict the alignment between sets of embeddings in one language and sets of embeddings in another. We show the model a set of 10-30 embeddings in English and another set of the same size of embeddings in French. The model is tasked to predict whether or not the embeddings in the two sets correspond to the same words.

Results on these tasks are shown in Table 3. Our model performed the highest across both tasks. Interestingly, the Single-Set Transformer and Union Transformer models performed quite well on the CoCo task (though not as well as our model), but were significantly worse on the FastText task. No other baseline aside from the Single-Set RN model performed notably better than chance. Given that the

| Model | CoCo | FastText |
|---|---|---|
| **Baselines** | | |
| PINE | 0.4977 ±0.0068 | 0.4977 ± 0.0024 |
| Single-Set RFF | 0.4964 ±0.0004 | 0.4974 ± 0.0028 |
| Single-Set RN | 0.7745 ±0.0365 | 0.7858 ± 0.0134 |
| Single-Set Transformer | 0.9064 ±0.0040 | 0.7698 ± 0.0114 |
| Union Transformer | 0.9285 ±0.0015 | 0.7319 ± 0.0163 |
| **Our Model** | | |
| Multi-Set Transformer | 0.9265 ±0.0128 | **0.8221** ± 0.0018 |
| Multi-Set RN | **0.9349** ±0.0189 | 0.7625 ± 0.0090 |
| Cross-Only | 0.9186 ±0.0119 | 0.8097 ± 0.0070 |
| Sum-Merge | 0.9303 ±0.0178 | 0.8160 ± 0.0092 |

Table 3: Average accuracy and standard deviation of each model across 3 runs on the alignment tasks.

| Model | Synthetic | Meta-Dataset |
|---|---|---|
| **Baselines** | | |
| PINE | 0.5012 ± 0.0020 | 0.5048 ± 0.0028 |
| Single-Set RFF | 0.5005 ± 0.0018 | 0.7831 ± 0.0069 |
| Single-Set RN | 0.4997 ± 0.0013 | 0.7981 ± 0.0642 |
| Single-Set Transformer | 0.6039 ± 0.0178 | 0.8811 ± 0.0092 |
| Union Transformer | 0.5908 ± 0.0057 | 0.7432 ± 0.0163 |
| **Our Models** | | |
| Multi-Set Transformer | 0.7289 ± 0.0353 | 0.8922 ± 0.0142 |
| Multi-Set RN | **0.7350** ± 0.0094 | 0.8679 ± 0.0111 |
| Cross-Only | 0.6353 ± 0.0191 | **0.9043** ± 0.0093 |
| Sum-Merge | 0.6292 ± 0.0101 | 0.8683 ± 0.0073 |

Table 4: Average accuracy and standard deviation of each model across 3 runs on the distinguishability tasks.

unaligned sets consisted of entirely disjoint images and captions (i.e., no images and captions overlapped), it's possible that learning whether the net sum of all embedded vectors in each set were aligned might be sufficient, and the model might not need to directly compare individual elements across sets. This might explain the high performance of the Single-Set Transformer and Union Transformer models - though interestingly, this did not hold true for the FastText task (perhaps due to the fact that the FastText task appeared to be more difficult).

### 5.2.3 Distinguishability

The last task in this category was distinguishability. Given two sets of samples, the models would be asked to predict whether the two input sets were drawn from the same underlying distribution. We again considered two examples for this task: a synthetic dataset, and a dataset of real-world images. For the synthetic data, we sampled sets from randomly generated 8-dimensional Gaussian mixtures (Gaussian mixture parameters were generated in the same fashion as the data in Section 5.1.1). For the second example, we used Meta-Dataset [Triantafillou et al., 2019][2], a dataset consisting of 12 other image datasets, each with many subclasses. In each case, each training example consisted of a batch of two sets of 10-30 data points. The data points would be drawn from the same distribution (the same Gaussian mixture for the synthetic data, or the same class from the same parent dataset for Meta-Dataset) with probability $1/2$, and generated from different distributions with probability $1/2$. The model was tasked to predict the probability of the sets being drawn from the same distribution. For the meta-dataset task, images were first encoded using a CNN trained along with the network.

Results are shown in Table 4. The Multi-Set Transformer

---

[2]Pytorch implementation taken from Boudiaf et al. [2021]

and Relation Network models performed by far the best on the synthetic task. On the image task, the Single-Set Transformer again performed very well, though not as well as the multi-set transformer model. We hypothesize this might be because the distinguishability task relies on recognizing the distribution from which the set is drawn, which is a task that might be possible to do by simply reducing each input set to a single vector and then comparing the resulting vectors.

### 5.3 ANALYSIS

Overall, we considered two different variants of our model (RN-based and transformer-based), as well as several ablations of the transformer model. The base multi-set transformer model performed consistently well across every task, with either the highest accuracy or close to the highest accuracy. The relation-network variant of the model performed slightly better on a number of tasks, but significantly worse on many others. This variant of the model also had significant issues with memory usage, and often required very small batch sizes in order to fit on GPUs. Each of the single-set transformer, cross-only, and sum-merge models can be considered ablations of the multi-set transformer architecture. The cross-only model performed competitively or slightly better on some tasks, but similar to the RN model, it performed worse by notable margins on others. It's possible that the $T_{XX}$ and $T_{YY}$ blocks - which computed the internal relationships between elements in $X$ and $Y$ - were simply not needed for certain tasks, but very helpful for others. The sum-merge model generally performed comparably to the base model, and degraded performance by notable margins only in the case of the distinguishability tasks. This is not entirely unexpected, given that it is the most minor of the ablations, and represents only a small change to the structure of the model.

## 5.4 SCALING

One important consideration when using set-based architectures is how the architectures will scale to large set sizes. Given input sets of size $n$ and $m$ and with model dimension $d$ (assuming that $d_{hidden}$ is of approximately the same order as $d_{latent}$), Table 5 shows the scaling properties of each model with the set sizes and latent dimension. The PINE and Single-Set RFF architectures are the fastest, scaling linearly with set size. All other models contain terms that are quadratic with set size, as they need to compare each element in one set to each element in another set (or the same set). Of these models, the transformer-based models (i.e. Single-Set Transformer, Union Transformer, Multi-Set Transformer, Cross-Only and Sum-Merge) all require approximately $(n+m)d^2 + (n+m)^2 d$ operations (though some need only $nmd$ or $(n^2+m^2)d$ due to omitting same-set terms or cross-set terms, the effect is still a net quadratic scaling with set size). The relation network models have the worst scaling properties, as they scale quadratically with the product of both set size and latent dimension. These scaling properties will remain the same if these architectures were generalized to $K > 2$ sets, with $n+m$ simply replaced by the total size of the union of all input sets.

While the PINE and Single-Set RFF architectures have the best scaling properties, they also demonstrate by far the worst performance, achieving results no better than chance on many of the image tasks. All of the transformer models share approximately the same scaling properties, scaling quadratically with the set sizes $n$ and $m$. This is a well-known property of transformer-based models, and a site of active research. Many previous works have proposed ways to reduce this quadratic dependency, and find approximations that allow these models to require only linear time (Wang et al. [2020], Choromanski et al. [2021],Kitaev et al. [2019], etc...). All of our proposed transformer models are compatible with any of these approaches, though we leave such explorations for future work. The Relation Network models are the most troublesome, as they scale quadratically with the product of both the model dimension and the set size. While these models do perform very well on some of the tasks discussed, they perform poorly on others, and overall the Multi-Set Transformer models exhibit both better scaling properties and more consistent performance across tasks.

## 6 DISCUSSION AND CONCLUSION

The Multi-Set Transformer model we define performs well at estimating a variety of distance/divergence measures between sets of samples, even for quantities that are notoriously difficult to estimate. It clearly outperforms existing multi-set and single-set architectures, and beats existing knn-based estimators in the settings we analyzed. In an

| Model | Ops. Scaling |
|---|---|
| PINE | $O\left((n+m)d^2\right)$ |
| Single-Set RFF | $O\left((n+m)d^2\right)$ |
| Single-Set RN | $O\left((n^2+m^2)d^2\right)$ |
| Single-Set Transformer | $O\left((n+m)d^2 + (n^2+m^2)d\right)$ |
| Union Transformer | $O\left((n+m)d^2 + (n+m)^2 d\right)$ |
| Multi-Set Transformer | $O\left((n+m)d^2 + (n+m)^2 d\right)$ |
| Multi-Set RN | $O\left((n+m)^2 d^2\right)$ |
| Cross-Only | $O\left((n+m)d^2 + nmd\right)$ |
| Sum-Merge | $O\left((n+m)d^2 + (n+m)^2 d\right)$ |

Table 5: Scaling of the number of operations required for each model with set sizes $n, m$ and dimension $d$.

ideal case, the model could be pretrained once and then applied as an estimator for, e.g., KL divergence in a diverse array of settings. This is one of our primary areas of focus going forward.

Since this model is a universal approximator for partially permutation equivariant functions, its applications are far broader than simply that of training estimators for divergences between distributions. We showcase a number of simple applications with image data, but these are merely meant to be representative of larger classes of applications. The applications in terms of distinguishability, for example, are highly reminiscent of GANs [Goodfellow et al., 2014], and the FastText task from the alignment section bears some similarities to existing work in which GANs are used [Lample et al., 2018], where our model might lead to improvements. Our model could also be used to train bespoke distance functions that could be trained end to end as part of a particular task. We hope to show these more diverse applications in greater detail in future work, as well as exploring better ways to train estimators that will generalize broadly, and looking at other quantities of interest such as Wasserstein Distance or f-divergences.

**Author Contributions**

Kira A. Selby and Pascal Poupart conceived the original idea. Kira A. Selby wrote the code, performed the experiments and wrote the paper. Ivan Kobyzev, Ahmad Rashid, Mehdi Rezagholizadeh and Pascal Poupart provided feedback and proofreading.

**Acknowledgements**

This research was funded by Huawei Canada and the National Sciences and Engineering Research Council of Canada. Resources used in preparing this research at the University of Waterloo were provided by the province of Ontario and the government of Canada through CIFAR and companies sponsoring the Vector Institute.

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
