# OpenReview forum: "Learning Functions on Multiple Sets using Multi-Set Transformers"
_auai.org/UAI/2022/Conference — UAI 2022 Poster_

### Official Review · Reviewer_u5Tm · 2022-04-09

**Q2(1) Originality/Novelty:** 3
**Q2(2) Significance/Impact:** 3
**Q2(3) Correctness/Technical Quality:** 3
**Q2(6) Clarity Of Writing:** 3
**Q6 Overall Score:** 5
**Q8 Confidence In Your Score:** 4

**Q1 Summary And Contributions:**

This paper proposes a general deep architecture for learning functions on multiple permutation-invariant sets. This architecture is further generalized to sets of elements of any dimension by dimension equivariance. Both theoretical analysis and experimental analysis are provided to show the effectiveness of the proposed architecture.

**Q2 Assessment Of The Paper:**

More detailed information regarding each of these aspects is given below:

**Q2(4) Quality Of Experiments (Optional):**

3: Good: The experimental evaluation is adequate, and the results convincingly support the main claims.

**Q2(5) Reproducibility:**

3: Good: Key resources (e.g., proofs, code, data) are available and key details (e.g., proofs, experimental setup) are sufficiently well-described for competent researchers to confidently reproduce the main results.

**Q3 Main Strengths:**

+ A general deep architecture for learning functions on multiple permutation-invariant sets is proposed.
+ Theoretical analysis shows the generalization ability of the proposed architecture.
+ Experimental analysis also shows the effectiveness of the proposed architecture in some ways.


**Q4 Main Weakness:**

- My major concern is on the insufficient comparison in Tables 2-3. Since Table 2 shows the results of multi-task image classification, many other representative multi-task learning methods should be included. Moreover, Table 3 shows the results of cross-modal learning in computer vision, and the latest multi-modal should be carefully discussed. At this moment, the reported results in Tables 2-3 are not that convincing.
- One important reference is missing, i.e., “Few-Shot Learning via Embedding Adaptation with Set-to-Set Functions, CVPR 2020”. Keeping this reference in mind, I wonder if the proposed architecture can be applied to few-shot learning.

**Q5 Detailed Comments To The Authors:**

See my detailed comments in Q4.

**Q7 Justification For Your Score:**

A general deep architecture for learning functions on multiple permutation-invariant sets is proposed. Both theoretical analysis and experimental analysis are provided to show the effectiveness of the proposed architecture. However, the comparisons in Tables 2-3 are insufficient.

**Q9 Complying With Reviewing Instructions:**

1: Yes.

---

### Official Review · Reviewer_A5Vt · 2022-04-09

**Q2(1) Originality/Novelty:** 3
**Q2(2) Significance/Impact:** 3
**Q2(3) Correctness/Technical Quality:** 3
**Q2(6) Clarity Of Writing:** 3
**Q6 Overall Score:** 7
**Q8 Confidence In Your Score:** 4

**Q1 Summary And Contributions:**

This paper extends previous work on single set transformer to handle inputs with multiple sets. Extensive experiments demonstrate the superiority of the proposed method.

**Q2 Assessment Of The Paper:**

More detailed information regarding each of these aspects is given below:

**Q2(4) Quality Of Experiments (Optional):**

3: Good: The experimental evaluation is adequate, and the results convincingly support the main claims.

**Q2(5) Reproducibility:**

2: Fair: Key resources (e.g., proofs, code, data) are unavailable but key details (e.g., proof sketches, experimental setup) are sufficiently well-described for an expert to confidently reproduce the main results.

**Q3 Main Strengths:**

1. the method is theoretically sound and straightforward.
2. they prove the method is an universal approximator on multiset functions.
3. the experiments are thorough and cover many different tasks.

**Q4 Main Weakness:**

1. the experiments are somewhat trivial. For example, the learned statistical distances are on Gaussian mixtures, which is questionable how good it can generalize to other distributions.
2. all experiments are supervised. It would be great to see some unsupervised applications.

**Q5 Detailed Comments To The Authors:**

The paper is well-written.

**Q7 Justification For Your Score:**

A novel method with sufficient experiments and significantly better results.

**Q9 Complying With Reviewing Instructions:**

1: Yes.

---

### Official Review · Reviewer_fQXr · 2022-04-10

**Q2(1) Originality/Novelty:** 2
**Q2(2) Significance/Impact:** 3
**Q2(3) Correctness/Technical Quality:** 4
**Q2(6) Clarity Of Writing:** 4
**Q6 Overall Score:** 7
**Q8 Confidence In Your Score:** 4

**Q1 Summary And Contributions:**

This paper presents a generalization of deep permutation invariant functions to multiple permutation invariant sets. They also extend their proposed method to sets of encode variable dimensions. The authors prove that their extended architecture is a universal approximator of continuous partially permutation equivariant function. The proposed method is then evaluated on multiple different tasks including alignment and statistical distance measurement.

**Q2 Assessment Of The Paper:**

More detailed information regarding each of these aspects is given below:

**Q2(4) Quality Of Experiments (Optional):**

4: Excellent: The experimental evaluation is comprehensive and the results are compelling.

**Q2(5) Reproducibility:**

3: Good: Key resources (e.g., proofs, code, data) are available and key details (e.g., proofs, experimental setup) are sufficiently well-described for competent researchers to confidently reproduce the main results.

**Q3 Main Strengths:**

-- The paper explores an interesting problem of extending previous work to sets of sets.

-- The idea of capturing different type of interaction using a separate transformer and concatenating to get a contextual representation of a set is novel (sec 3.3).

-- Showing that the resultant architecture is a universal approximator of partially permutation invariant function is simple and adequate. It would be better if in the main paper the authors could provide in the paper an outline of the proof and of where the is difficulty in proving 4.2. There is some white space that could be removed to get space for it, for example figure 1 can fit in single column.

-- I like the experiments section since they not only evaluate the performance of multi set transformer but they also provide ablation study of the different component. It is great to see that across all the application their proposed method is performing better than baselines, although there is no one clear winner.


**Q4 Main Weakness:**

-- The theoretical analysis shows that their proposed method is a universal approximator, but what is not clear is that their proposed architecture is the only way in which we can achieved that result.

-- Even though the ablation study is great, I would have liked to see how the method performs agains a simple baseline: T(S={X U Y}): R^d x R^d -> R^d'. Where the transformer looks at the entire set of X and Y. My guess is that in terms of performance this model will be at par with the best performing model.

**Q5 Detailed Comments To The Authors:**

-- If we try to extend this framework to 3 sets, 4 sets and beyond how many cross terms will be required in this given framework? Does this grow exponentially with the number of sets? Have you thought about such extension?

-- Topic models and other bag-of-bag of models have similar permutation invariances equivariances, have the authors tried experimenting with such models?



**Q7 Justification For Your Score:**

Overall the paper is solving an interesting problem with a novel architecture. The theoretical and empirical results shows that the proposed architecture does lead to better learning of partially equivariant functions. The paper is well written and easy to follow.

**Q9 Complying With Reviewing Instructions:**

1: Yes.

---

### Official Review · Reviewer_yjjC · 2022-04-13

**Q2(1) Originality/Novelty:** 2
**Q2(2) Significance/Impact:** 2
**Q2(3) Correctness/Technical Quality:** 3
**Q2(6) Clarity Of Writing:** 3
**Q6 Overall Score:** 6
**Q8 Confidence In Your Score:** 4

**Q1 Summary And Contributions:**

This paper deals with the general problem of neural architecture design in machine learning (ML). The authors here introduce a general deep architecture for learning functions on multiple permutation-invariant sets, and demonstrate that it is a universal approximator of these functions. Experimental comparisons show superior results to existing methods on a variety of ML tasks including counting tasks, alignment tasks, distinguishability tasks and statistical distance measurements.


**Q2 Assessment Of The Paper:**

More detailed information regarding each of these aspects is given below:

**Q2(4) Quality Of Experiments (Optional):**

2: Fair: The experimental evaluation is weak: important baselines are missing, or the results do not adequately support the main claims.

**Q2(5) Reproducibility:**

2: Fair: Key resources (e.g., proofs, code, data) are unavailable but key details (e.g., proof sketches, experimental setup) are sufficiently well-described for an expert to confidently reproduce the main results.

**Q3 Main Strengths:**

- Neural architecture design for a specific application is a difficult problem
- The paper is technically sound
- Experimentations are provided to support the model efficiency

**Q4 Main Weakness:**

- Experiment protocols are not identical for the 4 datasets
- Explanations of the type of algorithms (encoder, cnn, ...) used for experiments are missing
- Scalability of the proposal is not discussed. Only small size datasets are considered during experiments.



**Q5 Detailed Comments To The Authors:**

The concept proof of the proposal relies mainly on the experiments conducted. However, there is no information provided on the hyper-parameters of the compared approaches. How those hyper-parameters are optimized is not discussed.

Accuracy can be misleading, especially when dealing with imbalanced data. How does this affect the experimentations ? Critical difference diagram can also be useful to measure the statistical significance of the comparison between several algorithms.

Minors :
- Table 3 : Our ModelS
- Several references are incomplete.

**Q7 Justification For Your Score:**

The experimentations need improvement, wrt to the remarks above.

**Q9 Complying With Reviewing Instructions:**

1: Yes.

---

### Decision · Program_Chairs · 2022-05-15

**Decision:**

Accept (Poster)

**Comment:**

Meta Review: The authors present a new deep learning architecture for learning functions on multiple permutation-invariant sets. They also extend the architecture to handle variable dimensions. The paper presented both theoretical results that showing the architecture is a universal approximator of the functions, but also empirical results demonstrating nice performance on a number of benchmark tasks.  Overall the proposed method is novel, and the results are promising. The current architecture is illustrated using 2 sets. The authors explained that modeling k sets needs to model k^2 pairs of sets. Is that a limitation of the architecture? What about applying a set-invariant transformer on top of the sets?